# Mulvernet: Nucleus Segmentation and Classification of Pathology Images Using the HoVer-Net and Multiple Filter Units

**Vi Thi-Tuong Vo and Soo-Hyung Kim \***

Department of AI Convergence, Chonnam National University, Gwangju 61186, Republic of Korea
* Correspondence: shkim@jnu.ac.kr

**Abstract:** Nucleus segmentation and classification are crucial in pathology image analysis. Automated nuclear classification and segmentation methods support analysis and understanding of cell characteristics and functions, and allow the analysis of large-scale nuclear forms in the diagnosis and treatment of diseases. Common problems in these tasks arise from the inconsistent sizes and shapes of the cells in each pathology image. This study aims to develop a new method to address these problems based primarily on the horizontal and vertical distance network (HoVer-Net), multiple filter units, and attention gate mechanisms. The results of the study will significantly impact cell segmentation and classification by showing that a multiple filter unit improves the performance of the original HoVer-Net model. In addition, our experimental results show that the Mulvernet achieves outperforming results in both nuclei segmentation and classification compared to several methods. The ability to segment and classify different types of nuclei automatically has a direct influence on further pathological analysis, offering great potential not only to accelerate the diagnostic process in clinics but also for enhancing our understanding of tissue and cell properties to improve patient care and management.

**Keywords:** computational pathology; nucleus segmentation; nucleus classification; deep learning; multiple filter unit





## 1. Introduction

Pathology or histology images are stained by hematoxylin and eosin, allowing the efficient processing of tissues for analysis and management. Each pathology image contains tens of thousands of nuclei of different types, such as epithelial cells, inflammation, and neutrophils. These nuclei can be further analyzed to predict clinical outcomes, disease diagnosis, and prognosis. For example, nuclear features can be used to predict survival [1] and to diagnose disease type and grade [2]. Furthermore, efficient and accurate nucleus detection and segmentation can facilitate the quality of tissue segmentation [3,4] which, in turn, not only facilitates the quantification of pathology images, but also serves as an important step in understanding how each component of the tissue contributes to the disease. This necessitates the tasks of segmenting and classifying the nuclei in pathology analysis.

The segmentation and the classification of nuclei in previous work are separate processes. The available nuclei-segmentation methods are based on thresholds, image gradients, and morphological operations [5,6]. Recently, deep-learning methods have been widely used for segmenting nuclei [7,8] by predicting nucleus boundaries and using them for instance segmentation. Some have proposed using a nucleus distance map [9] or combining the nucleus distance map and the nucleus boundaries for nucleus segmentation [10]. In addition, ref. [11] integrated dense steerable filters into a convolutional neural network (CNN) to obtain the equivalence of rotation in nuclei segmentation.

The classification of nuclei has been studied in conjunction with the segmentation of nuclei. Many previous methods first segment individual nuclei and then classify them into ap-

propriate classes through quantitative features, such as intensity [12] morphology [12–14], and texture features [12], using machine-learning algorithms [12,14], and CNN [15,16]. Recently, ref. [17] proposed an end-to-end CNN horizontal and vertical distance network (HoVer-Net) for simultaneous nuclear instance segmentation and classification in multiple tissues.

However, nucleus segmentation and classification are challenging for several reasons. First, there are many nuclei of different sizes and shapes in a pathology image and it is difficult to analyze them manually. Second, tumor nuclei tend to cluster, resulting in complex contexts and many cases of overlapping.

Our main contributions are summarized as follows: (1) we propose a variant of HoVer-Net for improved simultaneous cell segmentation and classification; (2) we introduce the strategy to combine multiple filter units and attention gates into the original HoVer-Net in order to improve the performance of nucleus segmentation and classification; and (3) we demonstrate that the proposed method achieves outperforming performance on both nuclei segmentation and classification for diverse multi-tissue datasets.

The remainder of this paper is organized as follows. Section 2 describes the proposed Mulvernet and how it solves the problem of simultaneous cell segmentation and classification. The performance of cell segmentation and classification on various datasets is reported in Section 3. Finally, we discuss our experimental results in Section 4 and conclude the paper in Section 5.

## 2. Materials and Methods

This section introduces details of the proposed nuclei segmentation and classification model named Mulvernet, as shown in Figure 1. The input dimension is $256 \times 256$ and the output dimension of each branch is $164 \times 164$. First, the input is normalized and mapped to a 3-channel input form. Subsequently, the residual unit is used to extract a strong and representative set of features, due to the excellent performance of ResNet50 in recent computer vision tasks. Various residual units (RUs) are applied throughout the network at different downsampling levels. The order of the encoders is three RUs with down-sampling levels 1, four RUs with down-sampling levels 2, six RUs with down-sampling levels 4 and three RUs with down-sampling levels 8. The skip concatenation of the original HoVer-Net is replaced by an attention gate that can adjust the gain of the feature map to remove irrelevant and noisy responses in skip connections.

Following the encoder path, we perform three decoder paths for three subtasks: nuclei-classification task, horizontal and vertical distances of the pixel prediction task, and nuclei-segmentation tasks. All decoder paths utilize the same architectural design, consisting of a series of upsampling operations and multiple filter units. We use multiple filter units to incorporate low-level features and high-level features, which are particularly important in simultaneous tasks, where we aim for parallel nucleus segmentation and classification. The details of the residual unit and the multiple filter unit are also shown in Figure 1.

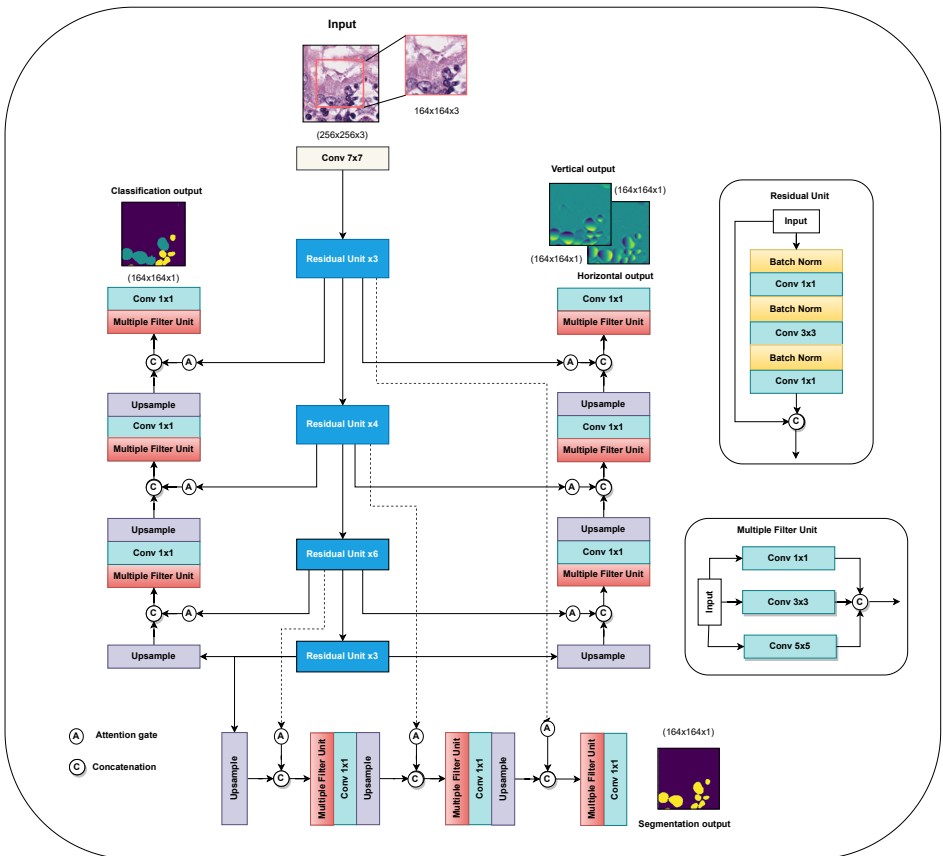

**Figure 1.** Schematic of our proposed method. The proposed method adopted to solve the problem, combining the HoVer-Net model [17] and a multiple filter unit [18], is named Mulvernet.

### 2.1. Multiple Filter Unit

The multiple filter unit [18] addresses the issue by increasing the filter size rather than iteratively reducing the image size. A multi-filter unit is a stack of three convolutional layers with different kernel sizes: $1 \times 1$, $3 \times 3$ and $5 \times 5$. Each filter learns different features. While small kernels extract small complex features, large kernels extract simpler features. Therefore, the size of the first convolution kernel filter is $1 \times 1$ to reduce the size of the input vector and extract local features. The next convolutional layer is a $3 \times 3$ convolution kernel using a downsampling size of 2 to obtain global features. The final convolutional layer has a kernel size of $5 \times 5$ and a downsampling size of 2. Then, all features are concatenated before proceeding to the next steps. Given an input $X$, the process of the multiple filter unit (MF unit) can be written as follows:

$$x' = max(0, F_{(X,f(1 \times 1))} \bigotimes F_{(X,f(3 \times 3))} \bigotimes F_{(X,f(5 \times 5))}), X \in R^{3 \times 256 \times 256} \tag{1}$$

where $F$ is the convolutional layer, $f$ is a filter of various sizes ($1 \times 1$, $3 \times 3$, and $5 \times 5$), $X$ is the feature map input of the multiple filter unit, $x'$ is the output of the multiple filter unit, and $\bigotimes$ represents the concatenation operation.

### 2.2. Attention Gate

Attention gates are integrated into the standard HoVer-Net architecture to focus on certain parts of the image and highlight relevant features that pass through skip connections. Given the feature map $X$ as an input and the gating signal $G \in R^{3 \times 256 \times 256}$ that is collected on a coarse scale and contains contextual information, the attention gate uses additive attention to obtain the gating coefficient. The input $X$ and the gating signal are first linearly mapped to an $R^{3 \times 256 \times 256}$ dimensional space, and then the output is squeezed into the

channel domain to produce a spatial attention weight map $S \in R^{3 \times 256 \times 256}$. The overall process can be written as follows:

$$S = \sigma(\varphi(\delta(\phi_x(X) + \phi_g(G))))$$ (2)

$$Y = SX$$ (3)

where $\varphi$, $\phi_x$ and $\phi_g$ are linear transformations implemented as $1 \times 1$ convolutions, $\delta$ is an element-wise nonlinearity, $\sigma$ is an activation function and Y is the output of the attention gate.

This attention gate is performed before the concatenation operation to combine only relevant activations and remove irrelevant and noisy responses in skip connections. In essence, this enables updates to the model parameters in shallow layers based mainly on spatial regions related to a specific task. This reduces computational resources wasted on unnecessary activation and improves the network generalization power.

Furthermore, we use Preact-ResNet50 [19] as the backbone.

## 3. Results

### 3.1. Datasets

We used three publicly available datasets to evaluate our method: the MoNuSAC, GlySAC, and CoNSeP datasets.

The MoNuSAC (multi-organ nucleus segmentation and classification) [20] dataset contains images from various organs, such as breasts, lungs, kidneys, and prostates. There are 209 images with 31,411 nuclei of four types: epithelial nuclei, lymphocytes, macrophages, and neutrophils. The training set contains 168 images, and the test set contains 41 images. Figure 2 shows some examples from the MoNuSAC dataset with the groundtruth and prediction by our proposed method.

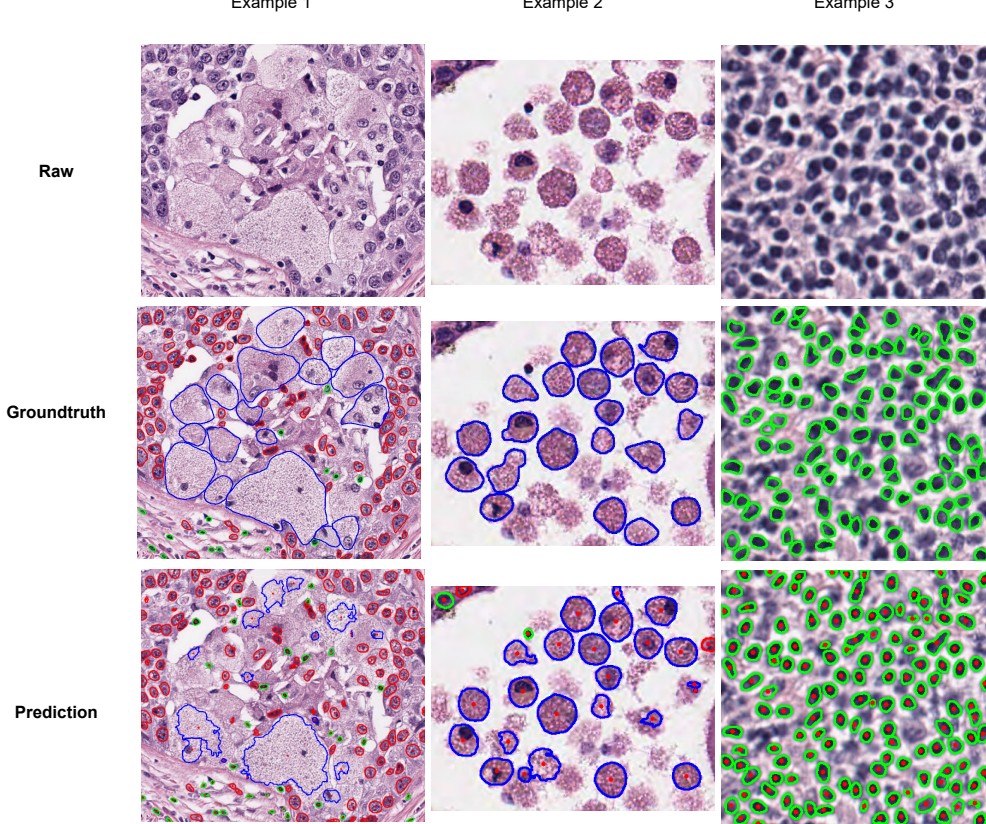

**Figure 2.** Sample tissue images from the MoNuSAC dataset with ground-truth annotations and prediction of Mulvernet model.

The GlySAC (gastric lymphocyte segmentation and classification) [21] dataset contains 59 images with various types of nuclei such as lymphocytes, cancerous epithelial and normal epithelial nuclei, stromal nuclei and endothelial nuclei. Sets of 34 and 25 images are used as the training and test sets, respectively. Figure 3 presents some examples from the GlySAC dataset with the groundtruth and prediction by Mulvernet.

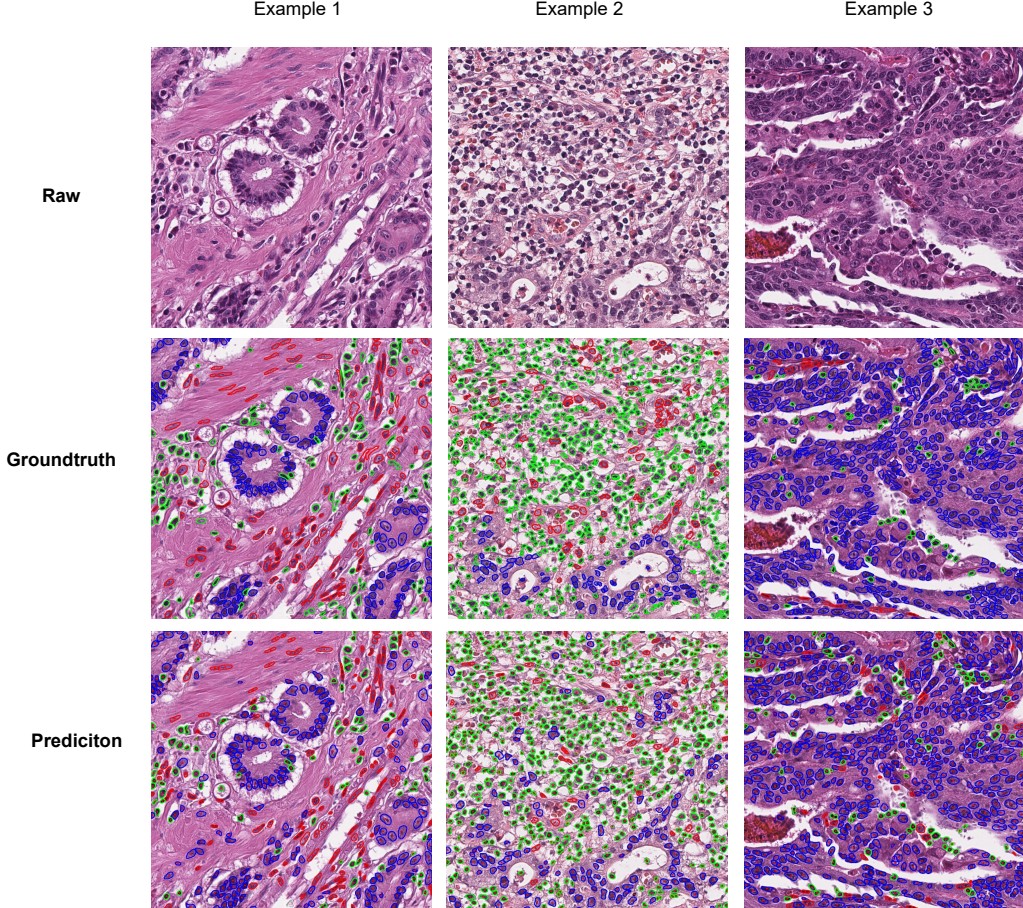

**Figure 3.** Sample tissue images from the GlySAC dataset with ground-truth annotations and prediction of Mulvernet model.

The CoNSeP (colorectal nuclear segmentation and phenotypes) [17] dataset contains 24,319 nuclei from 41 images. These nuclei comprise four types: miscellaneous, inflammatory, epithelial, and spindle. The CoNSep images are divided into a training set of 27 images and a test set of 14 images. Figure 4 shows some examples from the CoNSeP dataset with the groundtruth and prediction by Mulvernet.

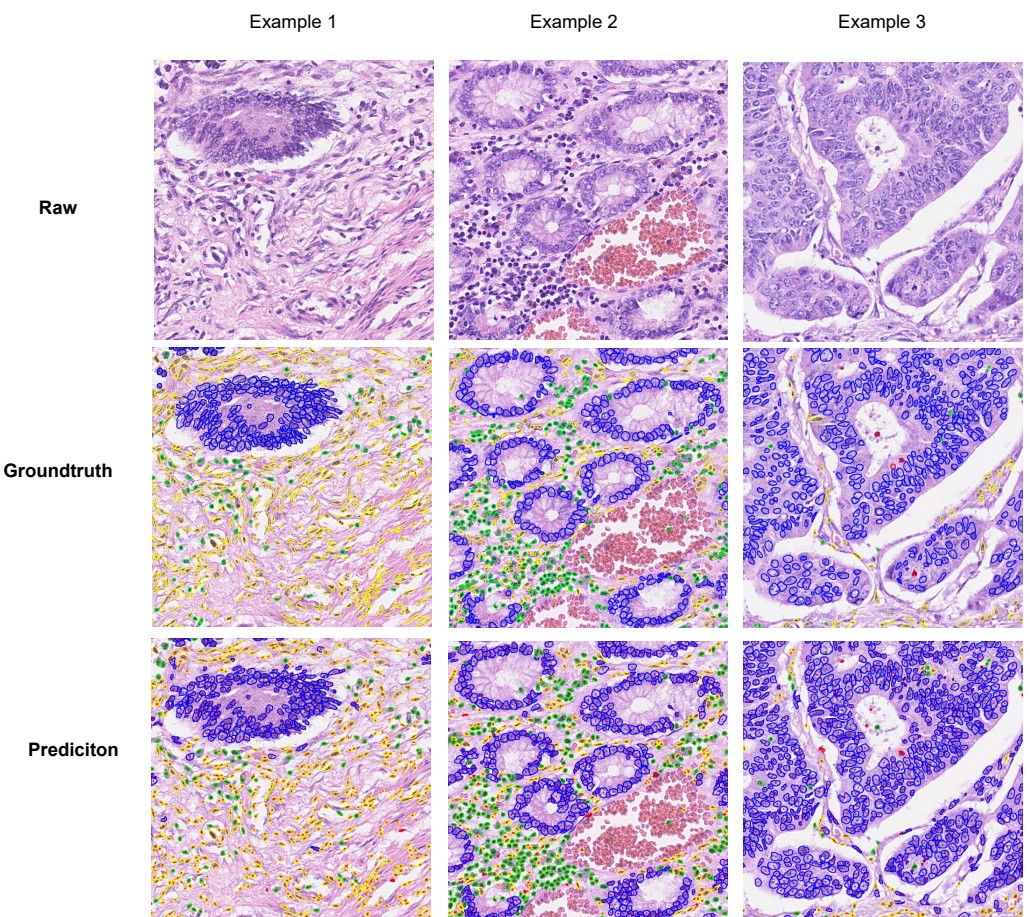

**Figure 4.** Sample tissue images from the CoNSeP dataset with ground-truth annotations and prediction of Mulvernet model.

*3.2. Evaluation Metrics*

We employed five evaluation metrics: dice score, aggregate Jaccard index [22], detection quality, segmentation quality, and panoptic quality [23]. Given the ground truth $x$ and the prediction $y$, the $TP$ computes the true positive, the $FP$ computes the false positive, $FN$ computes the false negative, and $IoU$ denotes the intersection over union of $x$ and $y$.

The dice score for measuring the separation of all nuclei from the background is defined as:

$$Dice = \frac{2 \times TP}{(TP + FP) + (TP + FN)} \tag{4}$$

Detection quality (DQ) is used to measure the instance detection, while the segmentation quality (SQ) evaluates how closely matched are predictions with ground truths. We formally define DQ and SQ as:

$$DQ = \frac{|TP|}{|TP| + 1/2|FP| + 1/2|FN|} \tag{5}$$

$$SQ = \frac{\sum_{((x,y) \in TP)} IoU(x,y)}{|TP| + \frac{1}{2}|FP| + \frac{1}{2}|FN|} \tag{6}$$

Panoptic quality (PQ) is proposed for nuclei instance segmentation [23]. PQ is aggregated from DQ and SQ components and is presented as follows:

$$PQ = DQ \times SQ \tag{7}$$

Aggregated Jaccard Index (AJI) is used to compute the nuclei-segmentation performance by computing the ratio of an aggregated intersection cardinality and an aggregated union cardinality between $x$ and $y$.

$$AJI = \frac{\sum_{i=1}^{N} |x_i \cap y_M^i|}{\sum_{i=1}^{N} |x_i \cup y_M^i| + \sum_{F \in U} |P_F|} \tag{8}$$

where $N$ is the number of nuclei, and $x_i$ and $y_i$ are the $i$th groundtruth and $i$th prediction, respectively. $y_M^i$ denotes the $M$th prediction which has the largest Jaccard Index with $x_i$. $U$ presents the connected component in the prediction without the corresponding ground truth.

AJI+ is the extension of AJI without over-penalization.

*3.3. Comparative Experiments*

Inspired by HoVer-Net, we propose a new method named Mulvernet. We compared our method through experiments with four representative models: NucleiSegNet [24], Triple U-Net [25], Mask-RCNN [26] and HoVer-Net.

NucleiSegNet [24] is based on the U-Net architecture, residual blocks and attention mechanisms to deal with the nuclei-segmentation problem without any post-processing step. Triple U-Net [25] uses the hematoxylin component to segment the nuclei. While NucleiSegNet [24] and Triple U-Net [25] were originally built for nuclei segmentation only, Mask-RCNN [26] was originally built for object localization and instance segmentation. HoVer-Net [17] is the horizontal and vertical distance network that is built for both nuclei segmentation and classification.

Table 1 presents the cell-classification results using the three datasets (MoNuSAC, GlySAC, and CoNSeP) and five methods. Our proposed method outperformed HoVer-Net and other competing models, regardless of the datasets.

**Table 1.** Cell-classification results using in three datasets (MoNuSAC, GlySAC, and CoNSeP) and five methods.

| Datasets | Method | Fd | F-Epithelial | F-Lymphocyte | F-Macrophages | F-Neutrophil |
|----------|--------|-----|--------------|--------------|---------------|--------------|
| MoNuSAC | NucleiSegNet [24] | 0.338 | 0.341 | 0.445 | 0.091 | 0.228 |
|  | Triple U-net [25] | 0.638 | 0.556 | 0.649 | 0.237 | 0.324 |
|  | Mask-RCNN [26] | 0.839 | 0.801 | 0.804 | 0.451 | 0.472 |
|  | HoVer-Net [17] | 0.825 | 0.754 | 0.803 | 0.382 | 0.387 |
|  | Proposed method | **0.841** | 0.764 | 0.829 | 0.371 | 0.435 |
|  |  | **Fd** | **F-Epithelial** | **F-Lymphocyte** | **F-Miscellaneous** |  |
| GlySAC | NucleiSegNet [24] | 0.712 | 0.369 | 0.429 | 0.115 | - |
|  | Triple U-net [25] | 0.728 | 0.401 | 0.463 | 0.106 | - |
|  | Mask-RCNN [26] | 0.818 | 0.513 | 0.535 | 0.279 | - |
|  | HoVer-Net [17] | 0.861 | 0.555 | 0.517 | 0.352 | - |
|  | Proposed method | **0.864** | 0.575 | 0.568 | 0.310 | - |
|  |  | **Fd** | **F-Epithelial** | **F-Inflammatory** | **F-Miscellaneous** | **F-Spindle** |
| CoNSeP | NucleiSegNet [24] | 0.418 | 0.310 | 0.216 | 0.098 | 0.288 |
|  | Triple U-net [25] | 0.632 | 0.358 | 0.561 | 0.102 | 0.438 |
|  | Mask-RCNN [26] | 0.731 | 0.608 | 0.598 | 0.099 | 0.516 |
|  | HoVer-Net [17] | 0.719 | 0.599 | 0.508 | 0.200 | 0.479 |
|  | Proposed method | **0.736** | 0.813 | 0.340 | 0.248 | 0.517 |

Table 2 compares the cell-segmentation performance of the five methods using five evaluation metrics and three datasets. Like the cell segmentation results, our proposed model was superior to the competing models on all datasets. The highest performance was in our proposed method with a dice score of 0.764, 0.835 and 0.833 on MoNuSAC, GlySAC and CoNSeP dataset, respectively. The lowest performance of the five methods

was the TripleUnet. Figure 5 shows the comparison between original HoVer-Net and Mulvernet through five evaluation metrics on three datasets. Overall, the Mulvernet achieved better performance than HoVer-Net. In other words, the effectiveness of multiple filter unit in improving the performance of HoVer-Net.

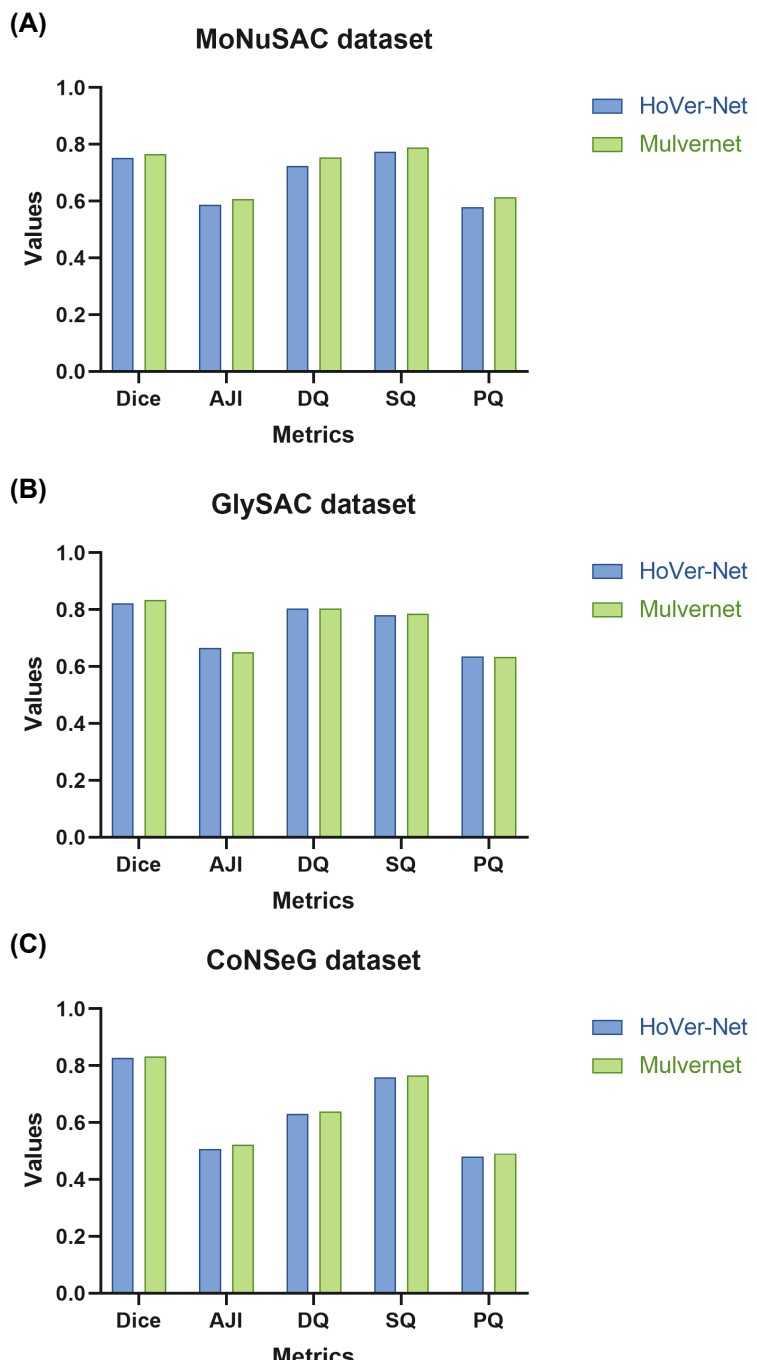

**Figure 5.** The comparison of five evaluation metrics in three datasets between original HoVer-Net and Mulvernet. (**A**) The comparison on MoNuSAC dataset. (**B**) The comparison on GlySAC dataset. (**C**) The comparison on CoNSeP dataset.

**Table 2.** Cell segmentation results on three datasets (MoNuSAC, GlySAC and CoNSeP) using Dice score.

| Datasets | NucleiSegNet [24] | Triple U-Net [25] | Mask-RCNN [26] | HoVer-Net [17] | Proposed Method |
|---|---|---|---|---|---|
| MoNuSAC | 0.537 | 0.512 | 0.767 | 0.753 | **0.766** |
| GlySAC | 0.651 | 0.677 | 0.781 | 0.823 | **0.835** |
| CoNSeP | 0.744 | 0.512 | 0.767 | 0.828 | **0.833** |

### 3.4. Ablation Experiments

The purpose of these ablation experiments was to assess the effectiveness of multiple filters. Tables 3 and 4 describe the results of the ablation experiments on cell classification and segmentation, respectively. Besides that, Figure 6 shows the comparison of the total loss in the validation sets of three datasets between the original HoVer-Net and Mulvernet. The effectiveness of multiple filters for classification is apparent. By combining three different filters, our proposed method improved the overall performance.

**Table 3.** Results of ablation experiments on cell classification.

| Datasets | Combinations | Fd | ACC | F-Epithelial | F-Lymphocyte | F-Macrophages | F-Neutrophil |
|---|---|---|---|---|---|---|---|
| MoNuSAC | $1 \times 1$ and $3 \times 3$ | 0.838 | 0.944 | 0.741 | 0.809 | 0.353 | 0.330 |
| | $1 \times 1$ and $5 \times 5$ | 0.833 | 0.953 | 0.754 | 0.813 | 0.340 | **0.517** |
| | $1 \times 1$ and $3 \times 3$ and $5 \times 5$ | **0.841** | **0.959** | **0.764** | **0.829** | **0.370** | 0.435 |

| | | Fd | ACC | F-Miscellaneous | F-Epithelial | F-Lymphocyte | |
|---|---|---|---|---|---|---|---|
| GlySAC | $1 \times 1$ and $3 \times 3$ | 0.861 | 0.709 | 0.297 | 0.552 | 0.549 | |
| | $1 \times 1$ and $5 \times 5$ | 0.863 | 0.713 | 0.285 | 0.564 | 0.556 | |
| | $1 \times 1$ and $3 \times 3$ and $5 \times 5$ | **0.864** | **0.725** | **0.310** | **0.575** | **0.568** | |

| | | Fd | ACC | F-Miscellaneous | F-Inflammatory | F-Epithelial | F-Spindle |
|---|---|---|---|---|---|---|---|
| CoNSeP | $1 \times 1$ and $3 \times 3$ | 0.733 | 0.768 | 0.204 | 0.459 | 0.571 | 0.430 |
| | $1 \times 1$ and $5 \times 5$ | 0.731 | 0.781 | 0.228 | **0.459** | 0.581 | 0.454 |
| | $1 \times 1$ and $3 \times 3$ and $5 \times 5$ | **0.736** | **0.784** | **0.248** | 0.340 | **0.813** | **0.517** |

**Table 4.** Results of ablation experiments on cell segmentation.

| Datasets | Filter Sizes | Dice | AJI | DQ | SQ | PQ | AJI+ |
|---|---|---|---|---|---|---|---|
| MoNuSAC | $1 \times 1$ and $3 \times 3$ | 0.745 | 0.589 | 0.717 | 0.779 | 0.579 | 0.593 |
| | $1 \times 1$ and $5 \times 5$ | 0.763 | 0.608 | **0.742** | 0.784 | 0.601 | 0.613 |
| | $1 \times 1$ and $3 \times 3$ and $5 \times 5$ | **0.766** | **0.608** | 0.737 | **0.789** | **0.601** | **0.613** |
| GlySAC | $1 \times 1$ and $3 \times 3$ | 0.835 | 0.647 | 0.799 | 0.786 | 0.629 | 0.661 |
| | $1 \times 1$ and $5 \times 5$ | 0.835 | **0.651** | 0.800 | **0.787** | 0.632 | 0.665 |
| | $1 \times 1$ and $3 \times 3$ and $5 \times 5$ | **0.835** | 0.650 | **0.804** | 0.786 | **0.634** | **0.666** |
| CoNSeP | $1 \times 1$ and $3 \times 3$ | 0.826 | 0.507 | 0.623 | 0.751 | 0.469 | 0.541 |
| | $1 \times 1$ and $5 \times 5$ | 0.825 | 0.486 | 0.610 | 0.746 | 0.456 | 0.514 |
| | $1 \times 1$ and $3 \times 3$ and $5 \times 5$ | **0.833** | **0.515** | **0.635** | **0.757** | **0.482** | **0.542** |

Similar results were seen for cell segmentation. The use of three filters boosted the segmentation performance. Therefore, by combining multiple filters, our proposed model facilitated the improved classification and segmentation of cells across all three datasets.

Table 3 shows the results of the cell-classification ablation experiments. The data were collected using three methods. Overall, a multiple filter unit with three sizes ($1 \times 1$, $3 \times 3$, and $5 \times 5$) performed best of all three combinations, with an $F_d$ of 0.841 on the MoNuSAC dataset, 0.864 on GlySAC dataset, and 0.740 on the CoNSeP dataset.

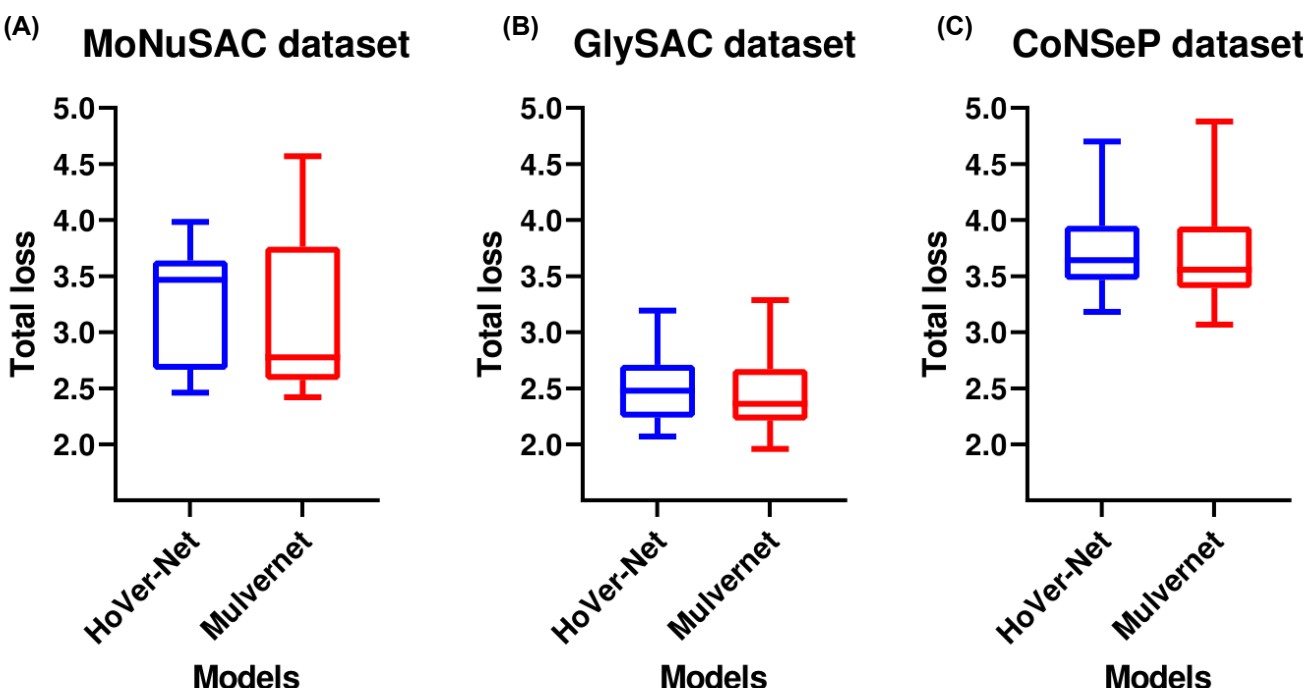

**Figure 6.** The comparison of total loss in the validation set of three datasets between original HoVer-Net and Mulvernet. (**A**) The comparison on MoNuSAC dataset. (**B**) The comparison on GlySAC dataset. (**C**) The comparison on CoNSeP dataset.

Table 4 compares the cell-segmentation results using six evaluation metrics according to three combinations of multiple filters on the MoNuSAC, GlySAC, and CoNSeP datasets. On the MoNuSAC dataset, the MF unit with filter sizes of $1 \times 1$ and $3 \times 3$ and $5 \times 5$ achieved the highest Dice score of 0.764, while the Dice score of filter sizes of $1 \times 1$ and $3 \times 3$ was 0.745. On the GlySAC and CoNSeP datasets, the Dice scores of the MF unit with filter sizes of $1 \times 1$ and $3 \times 3$ and $5 \times 5$ were 0.835 and 0.828, respectively, were higher than those of the other filter sizes combinations. Thus the ablation experiments demonstrated that the MF unit with filter sizes of $1 \times 1$ and $3 \times 3$ and $5 \times 5$ is the best choice for our proposed method.

*3.5. Implementation*

For network training, we used a patch as the input with a size of $256 \times 256$ pixels and a batch size of 4. For an input image, 0-1 normalization is applied before it is fed into the network. Furthermore, no data-augmentation technique was applied to this experiment. Our method is an end-to-end model. We trained the network using Adam optimization with a learning rate of $10^{-4}$ for 100 epochs until convergence. For loss computation, we calculated multiple regression losses, including the mean squared-error loss, cross-entropy loss [27], and dice loss [28] for simultaneous nucleus segmentation and classification was presented in [17]. Model selection was guided by the highest performance on the validation set. The Adam optimizer [29] was used as the optimization method for model training. All models were implemented using the PyTorch framework [30] with a NVIDIA GeForce 3090 Ti GPU (NVIDIA Corporate, Santa Clara, CA, USA).

**4. Discussion**

This study proposed a solution to the problem of parallel cell segmentation and classification in pathology images. We created a new simultaneous segmentation and classification model based on HoVer-Net combined with multiple filter units and attention mechanisms. We tested this method on the three datasets, assessed the performance of the segmentation and classification model, and compared it to several available methods.

The experimental results showed that our method achieved the best overall result. We also designed ablation experiments to demonstrate the effectiveness of the proposed model and combined the proposed module with the original HoVer-Net to analyze its effectiveness. The ablation-experiment results showed that the proposed multiple filters improved the performance of HoVer-Net to some extent. The above experimental results show that our method has some advantages in the automatic segmentation and classification of nuclei.

## 5. Conclusions

This paper presented a method for nucleus segmentation and classification from pathology images. Our method integrates multiple filter units into HoVer-Net with attention gates. The experimental results show the effectiveness of the multiple filter unit in improving the performance of the original HoVer-Net model as well as outperforming other models. The ability to segment and classify nuclei of different types automatically is directly associated with subsequent pathological analysis. It not only facilitates an excellent opportunity to speed up the diagnostic process in the clinic but also increases our understanding of tissue characteristics, leading to improved patient care and management. Our nucleus segmentation and classification model allows for the identification of morphological characteristics and quantification of the different types of nuclei and, thus, can provide additional diagnostic and predictive value. We observe low classification scores for nuclei with fewer samples and high variability. Future work will involve improving the class balance of data in simultaneous learning.

**Author Contributions:** Conceptualization, V.T.-T.V. and S.-H.K.; Methodology, V.T.-T.V. and S.-H.K.; Software, V.T.-T.V.; Validation, V.T.-T.V. and S.-H.K.; Formal analysis, V.T.-T.V. and S.-H.K.; Investigation, V.T.-T.V. and S.-H.K.; Resources, V.T.-T.V. and S.-H.K.; Data curation, V.T.-T.V. and S.-H.K.; Writing—original draft preparation, V.T.-T.V.; Writing—review and editing, S.-H.K.; Visualization, V.T.-T.V. and S.-H.K.; Supervision, S.-H.K.; Project administration, S.-H.K.; Funding acquisition, S.-H.K. All authors have read and agreed to the published version of the manuscript.

**Funding:** This research was supported by the Bio & Medical Technology Development Program of the National Research Foundation (NRF) and funded by the Korean government (MSIT) (NRF-2019M3E5D1A02067961) and also supported by Institute of Information & Communications Technology Planning & Evaluation (IITP) grant funded by the Korea government (MSIT) (No.2021-0-02068, Artificial Intelligence Innovation Hub).

**Data Availability Statement:** Not applicable.

**Conflicts of Interest:** The authors declare no conflict of interest.

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
