# Peer review of "Mulvernet: Nucleus Segmentation and Classification of Pathology Images Using the HoVer-Net and Multiple Filter Units"

_electronics, doi:10.3390/electronics12020355_

Round 1

Reviewer 1 Report

1. What is the main question addressed by the research?

Nucleus Segmentation and Classification of Pathology Images Using the HoVer-Net and Multiple Filter Units.

2. Do you consider the topic original or relevant in the field? Does it address a specific gap in the field? Accepted. The gaps have been addressed accordingly. However, it can be improved.   3. What does it add to the subject area compared with other published material? Yes.   4. What specific improvements should the authors consider regarding the methodology? What further controls should be considered? Please check my comments.   5. Are the conclusions consistent with the evidence and arguments presented and do they address the main question posed? Yes.   6. Are the references appropriate? Yes.

Further comments:

1. It is recommended to have at least five keywords for abstract to improve visibility.

2. The affiliation seems not right. Please include the country of the institution.

3. "However, nucleus segmentation and classification are challenging...". The problem statement is recommended to be written in the final paragraph in Introduction. Just before your research contribution.

4. Materials and Methods. The methodology must be explained in detail. It is recommended to explain the flowchart of methodology in order to improve readers understanding.

5. It is recommended to highlight the validation set. Three important things in machine learning: train, validation and test sets.

6. Must include hyperparameters.

7. It is recommended to briefly explain the compared four models.

8. Subsection 3.5 - The evaluation metrics must be stated before you explain the results.

9. Conclusion: Please include recommendation for future improvement.

10. Most of References are within five years. Acceptable.

Author Response

Firstly, we greatly appreciate your kind consideration of our manuscript. We have revised the manuscript in accordance with the comments of individual reviewers. We hope that the manuscript is now suitable for publication in the Electronics Journal.

Reviewer 2 Report

This manuscript proposes a variant of HoVer-Net with multiple filter units, termed Mulvernet, for nucleus segmentation and classification of pathology images, which just adds the multi-filter units with different kernel sizes to HoVer-Net and cannot be considered a new method. Furthermore, this study cannot give a new perspective. Some problems should be addressed as follows:  

  1. 1. In Section 1, the existing related work for nucleus segmentation and classification of pathology images is inadequate. What about the main contributions of this study?  

  1. 2. In Section 2, the authors proposed a method that combined a multiple filter unit with the Hover-Net model, moreover, the attention gate is also integrated into the Hover-Net. However, the theoretical presentation is weak, and equations (1) and (2) are irrelevant. In equation (1), please define the variable “n”. What about the “\sigma” in equation (2)? Is it an activation function? Please explain the relationship between x and X. The theoretical presentation for the combination of these three methods referring to Hover-Net, multiple filter units, and attention gates, is confusing. Furthermore, the innovation for the architecture is too small according to Figure 1. What about the first image in the first column in Figure 1?  

  1. 3. For Tables 2-4, please give the accurate names of the data sets. The evaluation metrics should be given before the experiments. Moreover, each variable and function should be given rigorous definitions, such as x, y, IoU, etc. AJI and AJI+ should be given the definitions and related equations.  

Author Response

(The authors gave the same response as above.)

Reviewer 3 Report

The paper is good and interesting, but it cannot be published in its current form. The abstract does not discuss why classification and segmentation of the nucleus are important to pathology analysis. Morever, the abstract does not discuss the result obtained. the author should improve those.

The limitation and strengths of the paper and directions for future work should be incorporated in the conclusion.

The organization of the paper should be included at the end of section 1 (the introduction).

Author Response

(The authors gave the same response as above.)

Round 2

Reviewer 2 Report

All my comments have been addressed.